# Parental breeding age effects on descendants' longevity interact over 2 generations in matrilines and patrilines

Zachariah Wylde[1]*, Foteini Spagopoulou[2], Amy K. Hooper[1], Alexei A. Maklakov[2,3], Russell Bonduriansky[1]

**1** Evolution & Ecology Research Centre, School of Biological, Earth and Environmental Sciences, University of New South Wales, Sydney, New South Wales, Australia, **2** Uppsala Centre for Evolution and Genomics, Uppsala University, Uppsala, Sweden, **3** School of Biological Sciences, University of East Anglia, Norwich Research Park, Norwich, United Kingdom

* wyldescience@gmail.com

**Data Availability Statement:** All data files are available from datadryad.org database DOI: https://doi.org/10.5061/dryad.2rbnzs7hw.

## Abstract

Individuals within populations vary enormously in mortality risk and longevity, but the causes of this variation remain poorly understood. A potentially important and phylogenetically widespread source of such variation is maternal age at breeding, which typically has negative effects on offspring longevity. Here, we show that paternal age can affect offspring longevity as strongly as maternal age does and that breeding age effects can interact over 2 generations in both matrilines and patrilines. We manipulated maternal and paternal ages at breeding over 2 generations in the neriid fly *Telostylinus angusticollis*. To determine whether breeding age effects can be modulated by the environment, we also manipulated larval diet and male competitive environment in the first generation. We found separate and interactive effects of parental and grand-parental ages at breeding on descendants' mortality rate and life span in both matrilines and patrilines. These breeding age effects were not modulated by grand-parental larval diet quality or competitive environment. Our findings suggest that variation in maternal and paternal ages at breeding could contribute substantially to intra-population variation in mortality and longevity.

## Introduction

In many species, offspring of older mothers have a reduced mean life span, a phenomenon known as the 'Lansing' effect [1] or maternal age effect. Maternal age effects have been observed in a great variety of organisms, including yeast, plants, nematodes, rotifers, insects, birds, and mammals [2–6]. Although most studies have focused on offspring life span, some studies show that maternal age at breeding can also affect offspring juvenile viability and adult reproductive performance [7–11]. A few studies have also reported effects of paternal age at breeding on offspring performance [2,5,6]. Parental age effects represent a potentially important source of variation in individual mortality risk, longevity, and fitness, but many aspects of these effects remain poorly understood.

**Funding:** https://www.arc.gov.au/ Future Fellowship FT120100274 and Discovery Grant DP170102449 awarded to RB. The funders had no role in study design, data collection and analysis, decision to publish, or preparation of the manuscript.

**Competing interests:** The authors have declared that no competing interests exist.

**Abbreviations:** AIC, Akaike Information Criterion; CHC, cuticular hydrocarbon; $F_1$, grand-parental generation; $F_2$, female and male offspring; $F_3$, grand-offspring; GSC, germline stem cell; HO, High condition Old; HY, High condition Young; KLDC, Kullback-Leibler discrepancy calibration; LMM, linear mixed models; LO, Low condition Old; LRT, likelihood ratio test; LY, Low condition Young; MCMC, Markov Chain Monte Carlo; RNAi, RNA interference; TERRA, telomeric repeat-containing RNAs.

Parental age effects could be caused by the accumulation of mutations in the germline [12]. In humans, mutations accumulate at a constant rate in the male germline and at an accelerating rate in the female germline [13]. Parental age effects could also be mediated by nongenetic factors. Recent studies on mice, monkeys, and humans have shown that patterns of DNA methylation across the genome change with age—a pattern known as the 'epigenetic clock' [14–18], and some of these altered epigenetic factors could be transmitted across generations [19–23]. Older parents could also transmit altered microRNAs or other factors such as proteins to offspring via the gametes [24,25]. For example, in mice, the transmission of proteins in the egg cytoplasm is thought to mediate maternal age effects on offspring [26], and more recent evidence suggests a role for sperm microRNAs in paternal effects [27–31]. Although such effects are best characterised in mammals, age-related changes in gamete quality also occur in arthropods, and such effects could contribute to parental age effects. For example, in the parasitoid wasp *Eupelmus vuilletti*, increasing maternal age is associated with reduced egg size and altered egg composition [32]. Likewise, in *Daphnia pulex*, maternal age is associated with changes in egg provisioning, with effects on offspring longevity and life history [33]. The transmission of dysregulated epigenetic or cytoplasmic factors from old-breeding parents to their offspring could mediate parental age effects in many species [34].

Maternal and paternal effects are likely to be mediated by different factors and can have distinct effects on offspring [35,36]. However, relatively few studies have tested experimentally for effects of paternal age at breeding, and even fewer studies have directly compared the effects of maternal and paternal age at breeding on offspring performance. Experimental evidence in mice shows that offspring of older fathers have a reduced life span and suggests that this effect could be mediated by epigenetic (DNA methylation) changes within sperm of gene promoters involved in evolutionarily conserved pathways of life span regulation [37]. In *Drosophila melanogaster*, both maternal and paternal age effects have been reported [5]. Similar effects may occur in other species (including humans), although much of the evidence is correlational. For example, in the wandering albatross, paternal but not maternal age affected juvenile survival of offspring [11]. A recent long-term study on a natural population of house sparrows showed that paternal breeding has a similar effect size on life span and reproductive success to female breeding age and that these effects are transferred to offspring in a sex-specific manner [6]. In humans, advanced paternal age at breeding is associated with reduced sperm quality and testicular functions, and such effects appear to be mediated by both epigenetic changes and genetic mutations [38]. Advanced paternal age is also associated with reduced performance on standardised tests in children, whereas the effect of maternal age was more complex [39]. Likewise, parental age, and the difference between maternal and paternal ages, are associated with risk of autism spectrum disorder [40].

Parental age effects could interact with environmental factors such as diet and stress [8,41]. For example, a restricted maternal diet mitigated the effects of advanced maternal age at breeding on offspring longevity in rotifers [42]. In mice, a fat-restricted maternal diet did not influence maternal age effects [16], but maternal age effects were mitigated by rapamycin [43]. In the butterfly *Pieris brassicae*, effects of parental age at breeding on offspring performance were influenced by stress [2]. However, the role of environment in modulating effects of parental age remains largely unexplored.

Perhaps the most important gap in understanding of parental age effects is the potential for such effects to accumulate and interact over multiple generations. In *Drospohila serrata*, offspring juvenile viability decreased with increasing maternal and grand-maternal ages at breeding [8], but it remains unclear whether such cumulative effects can occur in partrilines or in other species. If such multigenerational effects are widespread, they could make an important

contribution to variation in mortality and longevity and, potentially, play a role in the evolution of ageing [5,34].

Here, we examined 3 aspects of parental age effects that have received little attention in previous research by (1) comparing the effects of both male and female age at breeding on descendants, (2) testing for interactions of age at breeding with key environmental factors (diet and competitive environment), and (3) investigating the potential for effects of age at breeding to accumulate over generations. We addressed these questions in the neriid fly *Telostylinus angusticollis* (Enderlein), a species endemic to New South Wales and Southern Queensland, Australia. Both larval and adult nutrition affect mortality rate and life span in this species [44,45]. Larval access to dietary protein has a nonlinear effect on adult longevity [44], but high overall macronutrient (protein and carbohydrate) abundance at the larval stage accelerates larval growth and development while also promoting rapid ageing in males [46,47]. Adult protein restriction extends life [45] and can interact with larval diet to influence reproductive ageing [48]. However, effects of parental age at breeding on offspring performance have not been investigated previously in this species.

We reared individuals of the grand-parental ($F_1$) generation on either a high-nutrient or low-nutrient larval diet and then allowed adult females and males from these larval diet treatments to breed at 15 and 35 days of age. Neriid males fight other males for access to territories and females, and such male-male interactions could affect male ageing [47]. We therefore investigated the potential for male-male interactions to affect paternal age effects by manipulating $F_1$ male competitive environment. Female and male offspring ($F_2$) were reared on a standard larval diet (with a nutrient concentration intermediate between the high-nutrient and low-nutrient diets) and then allowed to breed at 15-day age intervals between ages 15 and 60 days. We quantified the adult longevity of grand-offspring ($F_3$) and used these data to test for effects of grand-parental ages at breeding, grand-parental environment, and parental ages at breeding on grand-offspring life span, mortality rate, and actuarial ageing rate.

## Results

### Life span

$F_3$ individuals (grand-offspring) from both matrilines and patrilines suffered similar negative effects of $F_1$ (grand-parental) and $F_2$ (parental) ages at breeding on life span (Table 1; Figs 1 and 2). $F_3$ individuals descended from old-breeding grandmothers and grandfathers had 37.8% and 39.8% shorter lifespans, respectively, than $F_3$ individuals descended from young-breeding grandmothers and grandfathers. There was no effect of $F_1$ larval diet on $F_3$ life span in either matrilines or patrilines, nor an $F_1$ larval diet × $F_1$ age interaction. There were also no main or interactive effects of $F_1$ male competitive environment within patrilines (S3 Table). However, we detected an $F_1$ × $F_2$ age interaction within both matrilines and patrilines, whereby the negative effect of $F_1$ age at breeding was diminished as $F_2$ age at breeding increased (Fig 2). Within matrilines, we also detected an interaction of $F_1$ age at breeding and $F_3$ sex, whereby the negative effect of grandmothers' age at breeding was stronger for $F_3$ males than for $F_3$ females. In patrilines, we also detected an $F_2$ age × $F_2$ sex interaction, such that $F_3$ life span declined more steeply with increasing paternal ($F_2$ male) age than with increasing maternal ($F_2$ female) age. S1 Fig shows the combined effects of $F_1$ and $F_2$ breeding ages, $F_1$ competitive environment (patrilines only), and $F_1$ larval diet on $F_3$ life span. Results were qualitatively similar for models including development time and body size (S4 Table). Overall, by comparison with previously published life span estimates for this species when maintained as individually housed virgin adults (e.g., male median = 37 d, female median = 36 d; [49]), the median lifespans of $F_3$ individuals descended from young-breeding parents and grandparents

**Table 1. Tests of effects based on linear mixed models of $F_3$ life span for patrilines and matrilines.** Significant effects are highlighted in bold. Negative effects of $F_1$ and $F_2$ age indicate that old grandparents and parents produced $F_3$ individuals with reduced lifespans, negative effects of larval diet indicate that low-nutrient larval diet has a negative effect on $F_3$ life span, and negative effects of sex indicate that the life span of male descendants was lower than that of females. Effect sizes represent marginal $R^2$. Conditional whole-model $R^2$ values were 47.72% for the patriline model and 54.78% for the matriline model.

| Effects on $F_3$ life span | Patrilines | | | | | | Matrilines | | | | | |
|---|---|---|---|---|---|---|---|---|---|---|---|---|
| Fixed effects: | Estimate | SE | F | $X^2$ | P | Effect size (%) | Estimate | SE | F | $X^2$ | P | Effect size (%) |
| (Intercept) | 81.958 | 6.956 | – | 138.809 | <0.001 | – | 91.294 | 6.624 | – | 189.944 | <0.001 | – |
| $F_1$ larval diet | −8.504 | 4.619 | 2.620 | 3.389 | 0.066 | 0.258 | −6.437 | 4.182 | 1.700 | 2.369 | 0.124 | 2.97 |
| $F_1$ age | −22.325 | 5.247 | 20.227 | 18.106 | <0.001 | 30.8 | −20.256 | 4.414 | 25.154 | 21.058 | <0.001 | 35.38 |
| $F_2$ sex | 8.321 | 5.374 | 1.316 | 2.397 | 0.122 | 5.26 | 1.566 | 4.980 | 0.428 | 0.099 | 0.753 | 0.030 |
| $F_2$ age | −0.948 | 0.155 | 40.983 | 37.404 | <0.001 | 15.45 | −1.177 | 0.157 | 46.448 | 56.317 | <0.001 | 35.62 |
| $F_3$ sex | −17.846 | 4.359 | 16.712 | 16.759 | <0.001 | 10.75 | −32.761 | 4.551 | 45.070 | 51.818 | <0.001 | 39.55 |
| $F_1$ age × $F_2$ age | 0.266 | 0.112 | 5.606 | 5.606 | 0.018 | 10.85 | 0.254 | 0.102 | 6.249 | 6.249 | 0.012 | 11.33 |
| $F_1$ larval diet × $F_1$ age | 3.482 | 3.460 | 1.013 | 1.013 | 0.314 | 1.31 | 0.793 | 2.518 | 0.099 | 0.099 | 0.753 | 0.0511 |
| $F_1$ larval diet × $F_2$ sex | −1.533 | 3.011 | 0.259 | 0.259 | 0.611 | 0.181 | −1.068 | 2.605 | 0.168 | 0.168 | 0.682 | 0.090 |
| $F_1$ age × $F_2$ sex | −0.438 | 3.222 | 0.019 | 0.019 | 0.892 | 0.0151 | −1.022 | 2.621 | 0.152 | 0.152 | 0.697 | 0.100 |
| $F_2$ sex × $F_2$ age | −0.205 | 0.103 | 3.957 | 3.957 | 0.047 | 4.29 | −0.153 | 0.092 | 2.758 | 2.758 | 0.097 | 2.9 |
| $F_1$ age × $F_3$ sex | 3.796 | 2.732 | 1.931 | 1.931 | 0.165 | 1.55 | 5.361 | 2.362 | 5.150 | 5.150 | 0.023 | 2.99 |
| $F_2$ sex × $F_3$ sex | −3.549 | 2.614 | 1.843 | 1.843 | 0.175 | 0.899 | 4.209 | 2.420 | 3.026 | 3.026 | 0.082 | 1.64 |
| $F_2$ age × $F_3$ sex | 0.120 | 0.079 | 2.351 | 2.351 | 0.125 | 2.44 | 0.425 | 0.080 | 28.587 | 28.587 | <0.001 | 25.33 |
| $F_1$ larval diet × $F_3$ sex | 4.482 | 2.494 | 3.230 | 3.230 | 0.072 | 1.94 | 3.741 | 2.384 | 2.463 | 2.463 | 0.117 | 1.21 |

are similar (male median = 25, female median = 36), whereas the median lifespans of $F_3$ individuals descended from old-breeding parents and grandparents are substantially lower (male median = 10, female median = 15).

## Mortality rate

Consistent with our results for life span, we found that baseline mortality rate (Gompertz $b_o$ parameter) of $F_3$ individuals from both matrilines and patrilines was affected positively and

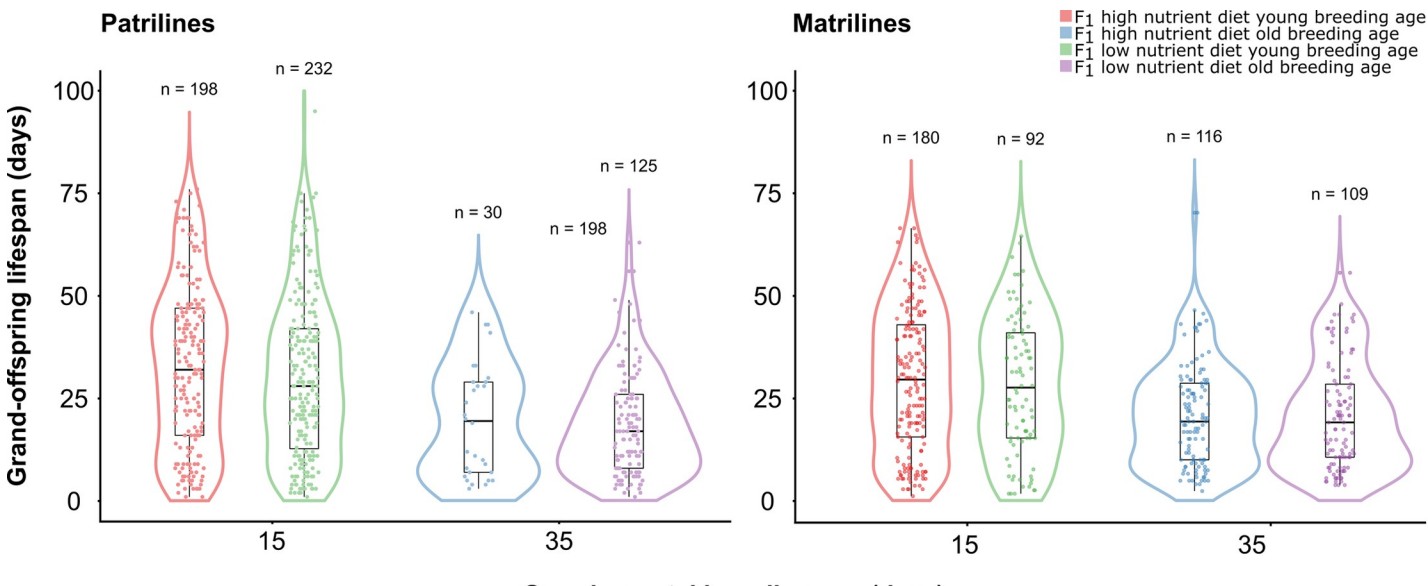

**Fig 1. Effects of grand-parental ($F_1$) breeding age and larval diet on grand-offspring ($F_3$) life span in patrilines and matrilines.** The violin plot outline illustrates kernel probability density (width represents proportion of data located there). Within violin plots are box plots with median and interquartile range to illustrate data distribution. Underlying data can be found in the Dryad Repository: https://doi.org/10.5061/dryad.2rbnzs7hw.

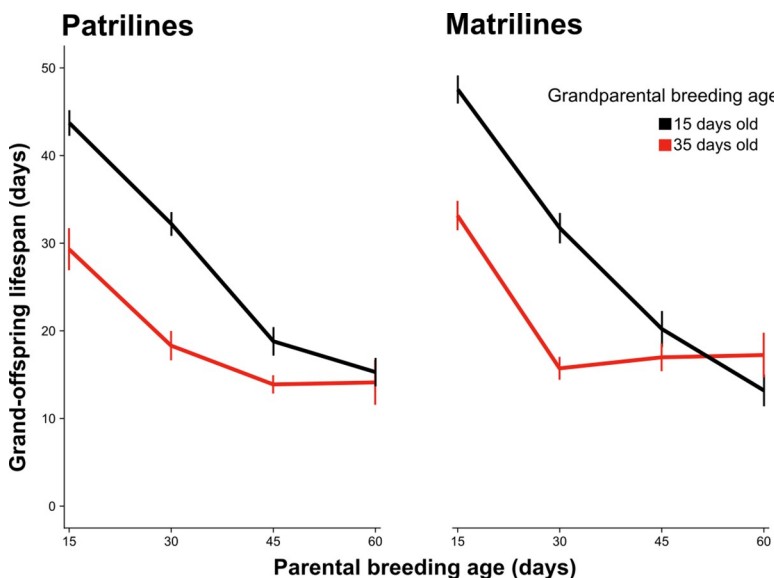

**Fig 2. Interaction between effects of grand-parental and parental breeding ages on grand-offspring life span in patrilines and matrilines.** Black lines represent the lifespans of $F_3$ descendants of $F_1$ individuals paired at 15 days of age, and red lines represent the lifespans of $F_3$ descendants of $F_1$ individuals paired at 35 days of age. Bars represent SEM. Underlying data can be found in the Dryad Repository: https://doi.org/10.5061/dryad.2rbnzs7hw. $F_1$, grand-parental generation; $F_3$, grand-offspring.

similarly by $F_1$ age at breeding but not affected by $F_1$ larval diet (Fig 3). Individuals descended from grandparents that bred at age 35 d had higher baseline mortality rates, regardless of $F_1$ larval diet treatment (High condition Old [HO]; Low condition Old [LO]; patrilines $b_{0\ HO} = -3.5$, $b_{0\ LO} = -3.6$; matrilines $b_{0\ HO} = -3.8$, $b_{0\ LO} = -3.7$) than individuals descended from grandparents that bred at age 15 d (High condition Young [HY]; Low condition Young [LY]; patrilines $b_{0\ HY} = -4.4$, $b_{0\ LY} = -4.2$; matrilines $b_{0\ HY} = -4.6$, $b_{0\ LY} = -4.4$). An effect of $F_1$ age at breeding on the baseline mortality rate was supported by Kullback-Leibler discrepancy calibration (KLDC) values, which exceeded 0.98 for all comparisons of $b_0$ parameters for $F_3$ descendants of young-breeding versus old-breeding $F_1$ individuals within and across larval diet treatments in both patrilines and matrilines (S6 and S8 Tables).

Grand-parental and parental breeding ages interacted in their effects on $F_3$ baseline mortality rates ($b_0$), particularly within patrilines (Fig 4). $F_3$ individuals descended from young grandparents ($F_1$) experienced increasingly high baseline mortality as parental ($F_2$) age at breeding increased, and this effect was especially strong in patrilines (S10 and S11 Tables). By contrast, for $F_3$ individuals descended from old-breeding grandparents, there were no consistent effects of parental age at breeding.

For actuarial ageing rates (Gompertz $b_1$ parameter), evidence of treatment effects was weaker, and patterns were less consistent. Individuals descended from grandparents that bred at age 35 days had similar actuarial ageing rates, regardless of $F_1$ larval diet treatment (patrilines $b_{1\ HO} = 0.032$, $b_{1\ LO} = 0.036$; matrilines $b_{1\ HO} = 0.031$, $b_{1\ LO} = 0.029$), to individuals descended from grandparents that bred at age 15 d (patrilines $b_{1\ HY} = 0.032$, $b_{1\ LY} = 0.029$; matrilines $b_{1\ HY} = 0.035$, $b_{1\ LY} = 0.034$). In matrilines, KLDC values were $<0.85$ for all comparisons of $b_1$ parameters for $F_3$ descendants of young-breeding versus old-breeding $F_1$ females (S8 Table). In patrilines, KLDC values marginally exceeded 0.85 for some comparisons of $F_3$ descendants of young-breeding versus old-breeding $F_1$ males within and across larval diet treatments, but the effect of $F_1$ age at breeding on $b_1$ was not consistent across larval diet

# Patrilines

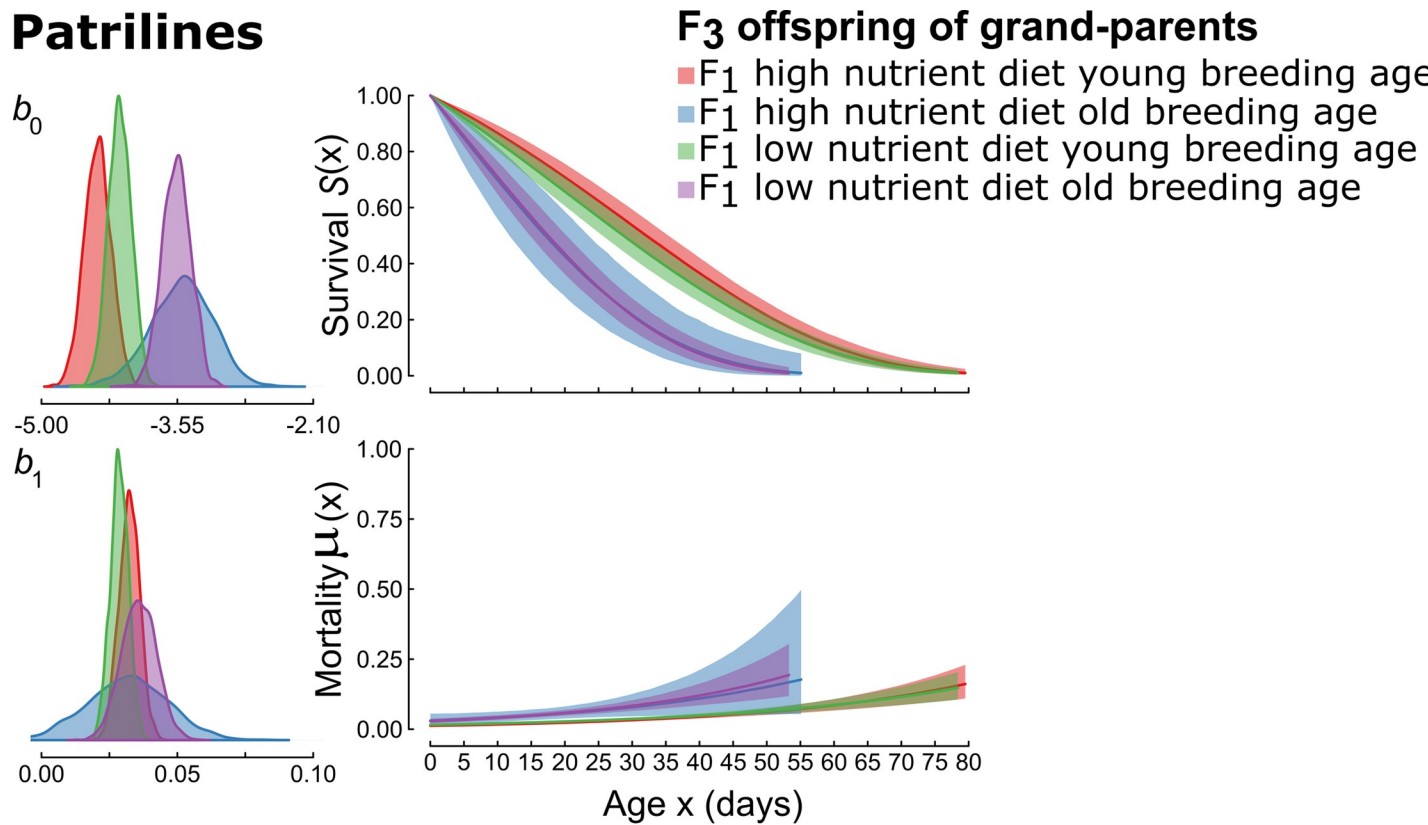

# Matrilines

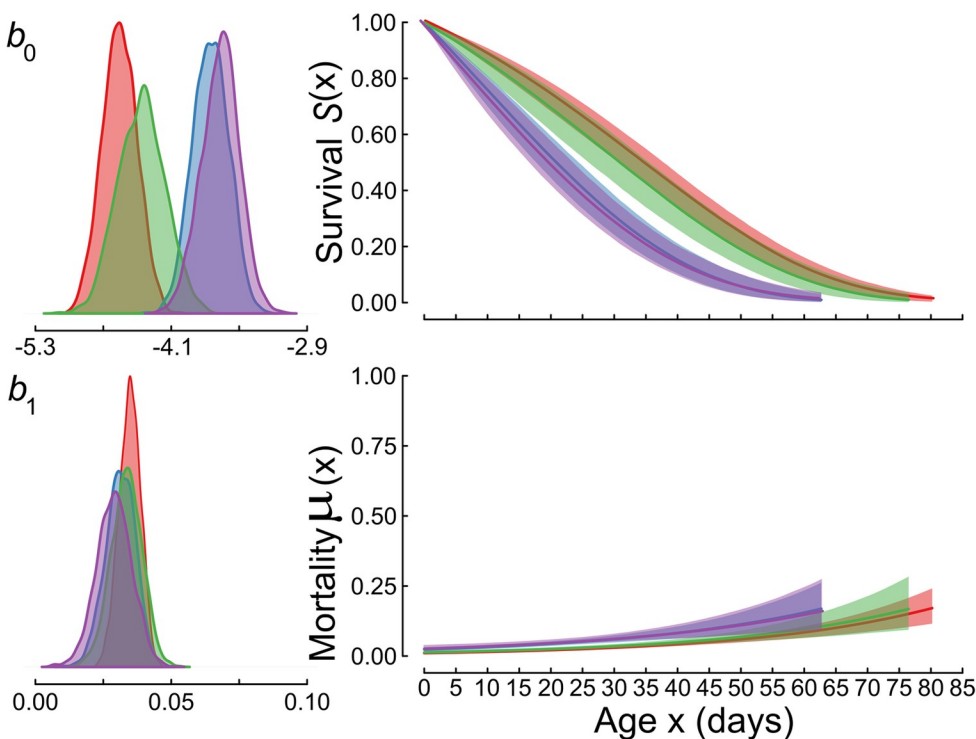

**Fig 3. Effects of grand-parental larval diet and breeding age on estimated age-specific survival and mortality rates for grand-offspring of patrilines and matrilines as fitted by the simple Gompertz mortality model.** $b_0$ is the baseline mortality rate (scale) parameter, and $b_1$ is the rate of actuarial ageing (shape) parameter. Posterior distributions are shown for $b_0$ and $b_1$ in the left panels. Panels on the right illustrate how these estimates translate to survival and mortality rates over time. The shaded areas in the survival plots represent 95% confidence intervals. Underlying data can be found in the Dryad Repository: https://doi.org/10.5061/dryad.2rbnzs7hw. $F_1$, grand-parental generation; $F_3$, grand-offspring.

treatments (S7 Table). There was little evidence that grand-parental and parental ages at breeding interacted in their effects on actuarial ageing rate ($b_1$) in either matrilines or patrilines (Fig 4). Wider confidence limits for life span and age-dependent mortality rates for descendants of old-breeding $F_1$ males reflect reduced sample size resulting from mortality between 15 and 35 days of age. For all other KLDC values of group comparisons refer to S2–S5 Figs and S9–S13 Tables.

## Discussion

A recent model suggests that negative effects of parental age on offspring performance can readily evolve [50], but many aspects of such effects have received little attention in empirical research. Our results show that paternal age effects can be similar in magnitude to maternal age effects. The magnitude of the grand-maternal and grand-paternal effects detected in our study is comparable to longevity changes observed in multigenerational selection experiments in *Drosophila melanogaster* [51,52]. Our mortality rate analyses suggest that decreased life span of grand-offspring of older grandparents and parents results largely from elevated baseline mortality rather than from a higher rate of increase in mortality rate with age (i.e., actuarial ageing). Actuarial ageing could result from the accumulation of somatic damage with age [53]. Previous studies of *T. angusticollis* showed that males reared on a high-nutrient larval diet accumulated damage more rapidly with age than males reared on a low-nutrient larval diet [46] and exhibited more rapid actuarial and reproductive ageing [47]. Here, we show that declining offspring longevity and increasing offspring mortality rate represent additional manifestations of ageing in *T. angusticollis* males and females. However, breeding age effects on offspring life span and mortality were unaffected by grand-parental larval diet. Interestingly, although we found largely similar effects of grand-paternal versus grand-maternal and paternal versus maternal ages at breeding on offspring baseline mortality rate, we also found some evidence of effects on actuarial ageing rate in patrilines but not in matrilines. These differences suggest that male and female breeding age effects could be mediated by different factors and could have different effects on offspring life history.

Our findings suggest that the effect of ancestors' age at breeding could contribute substantially to within-population variation in longevity. However, the importance of these effects in natural populations remains unclear. *T. angusticollis* has a much shorter mean life span in the wild than in the laboratory, and wild males also exhibit very rapid actuarial ageing [49]. The short average life span and rapid ageing observed in natural populations of this species is consistent with findings for other insects in the wild [54–56]. Given the very high background mortality rate experienced by *T. angusticollis* in the wild, it is possible that longevity of flies in natural populations is not strongly affected by parental age effects. However, it is also possible that maternal and paternal age effects are accelerated along with the overall rate of ageing in wild populations as a result of environmental stresses such as parasites and temperature fluctuations. If so, then parental age effects could have a substantial effect on fitness in natural populations, despite a short life expectancy. It is also possible that offspring of old-breeding parents or grandparents might respond by increasing their early-life reproductive effort, thereby partly mitigating the effects of reduced life span. For example, in *Daphnia* pulex, older mothers produce offspring with shortened life spans but these offspring achieve increased early-life

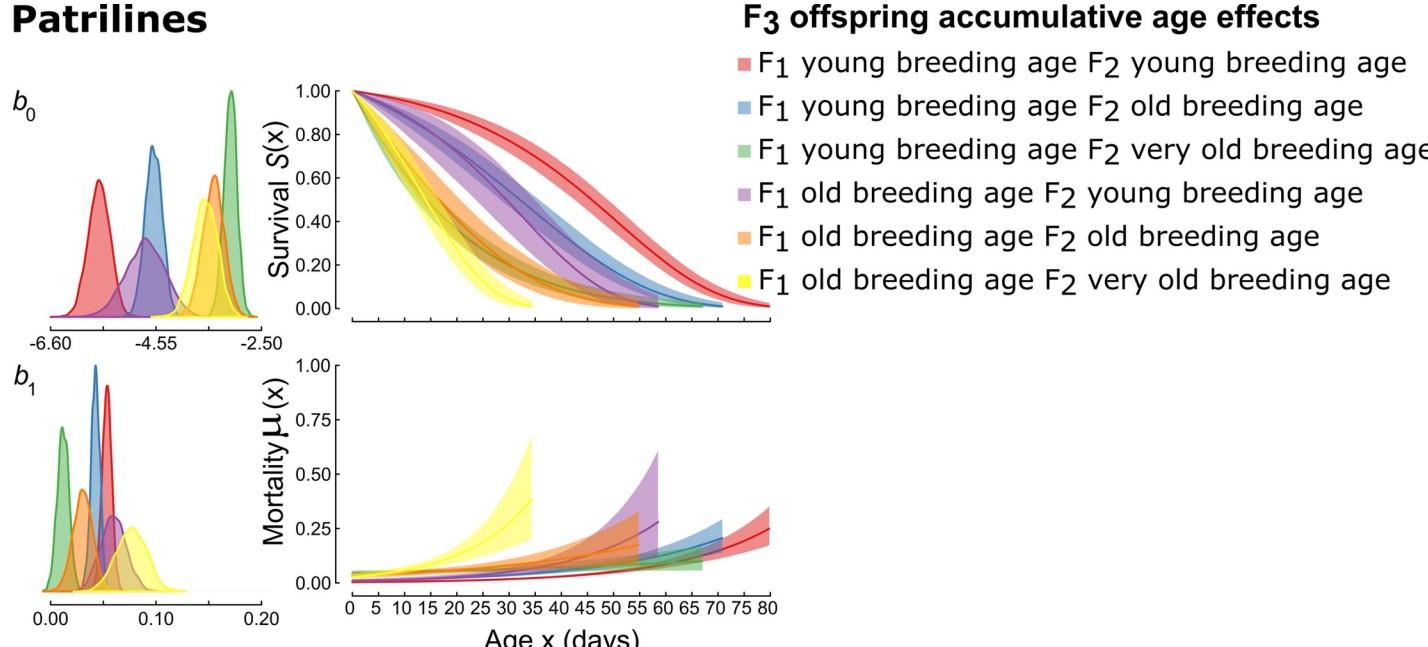

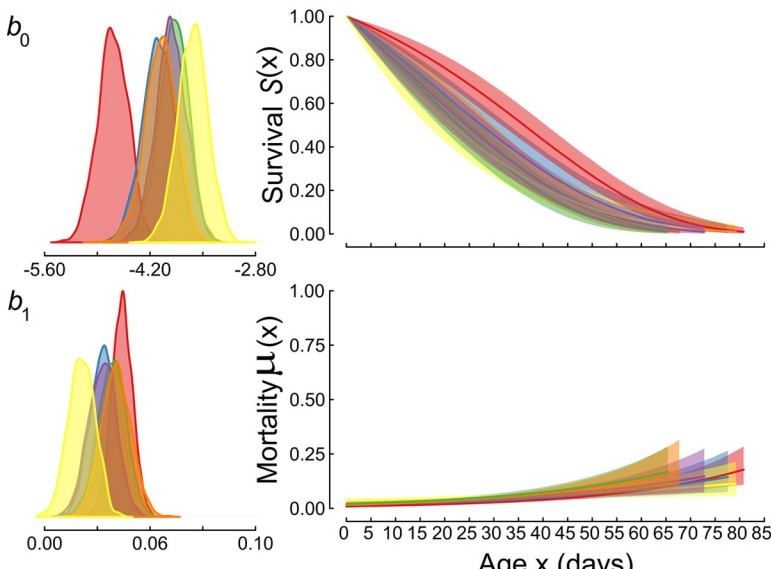

**Fig 4. Effects of $F_1$ breeding age and $F_2$ breeding age on estimated age-specific survival and mortality rates for grand-offspring of patrilines and matrilines as fitted by the simple Gompertz mortality model.** $b_0$ is the baseline mortality rate (scale) parameter, and $b_1$ is the rate of actuarial ageing (shape) parameter. Posterior distributions are shown for $b_0$ and $b_1$ in the left panels. Panels on the right illustrate how these estimates translate to survival and mortality rates over time. Shaded areas in the survival plots represent 95% confidence intervals. Underlying data can be found in the Dryad Repository: https://doi.org/10.5061/dryad.2rbnzs7hw. $F_1$, grand-parental generation; $F_2$, female and male offspring; $F_3$, grand-offspring.

reproductive output [33]. We found little evidence that age at breeding effects on life span were mediated by body size or development time, because inclusion of these traits as covariates in life span models did not qualitatively alter the results.

The grand-parental and parental age effects that we observed could be mediated by the accumulation of germline mutations with age. Because male and female germline cells develop differently in animals, including flies [57–59], the male and female germlines could accumulate mutations at different rates [60,61]. In particular, the rate of age-dependent mutation accumulation is likely to reflect the number of germline cell divisions, and it has long been thought that males transmit more germline mutations because the male germline undergoes a larger number of cell divisions [62]. Interestingly, however, in *Drosophila*, the number of germline cell divisions is larger in females than in males at young ages but larger in males than in females at old ages [63]. This suggests that mutation-mediated maternal and paternal age effects could differ in relative magnitudes as a function of male and female age. If *T. angusticollis* exhibits a similar pattern of germline cell division to *Drosophila*, this could explain the somewhat stronger negative effect of grand-paternal age at breeding on grand-offspring life span, relative to the effect of grand-maternal age at breeding (Fig 1).

The rate of cell proliferation in the female germline also increases on a protein-rich diet in *D. melanogaster* [64], and dietary protein strongly stimulates female fecundity in *T. angusticollis* as well [45]. A protein-rich adult diet could therefore be expected to accentuate negative maternal breeding age effects on offspring performance and could also accentuate paternal breeding age effects if cell division in the male germline is also enhanced on a high-protein diet. Germline mutation rate can also be affected by investment in DNA repair, and *D. melanogaster* reared on low-nutrient food as larvae have lower rates of repair that result in increased germline mutation rate [65]. However, we found little evidence of effects of $F_1$ larval diet on grand-offspring mortality and survival (Figs 2 and 4). Likewise, we did not detect an effect of male competitive environment (opportunity for combat interactions) or any interaction between this treatment and grand-paternal breeding age. This finding is consistent with the lack of any effect of male combat on male reproductive ageing [47] and suggests that agonistic interactions with other males do not affect the maintenance of the male germline.

A different (but nonexclusive) explanation for our findings is age-dependent transmission of epigenetic or cytoplasmic factors through the female and male germlines. DNA (cytosine) methylation contributes to the regulation of gene expression in many organisms [66], but flies have little cytosine methylation and its role in this group remains unclear [67–70]. In *D. melanogaster*, DNA methylation is largely limited to the early stages of embryogenesis [71,72], but 2 studies suggest that DNA methylation can also persist in the germline [73,74]. In mammals, DNA methylation patterns undergo changes with age throughout the genome [75,76]. Such age-related changes in methylation (known as the 'epigenetic clock') could mediate parental age effects, because some DNA methylation patterns can be transmitted to offspring via both sperm and eggs (for a review, see the work by Ho and Burggren [77]). It is not known whether a DNA methylation 'clock' also occurs in flies.

Other epigenetic or cytoplasmic factors that change with age could also mediate the observed age-at-breeding effects. There is evidence of age-related cellular changes in the male and female germline. For example, as *Drosophila* males age, germline stem cells (GSCs) divide less frequently because of misorientation of centromeres [78]. Similarly, GSC division in female *Drosophila* declines with age, and this is accompanied by an increased rate of cell death in developing eggs [79]. RNA-mediated transmission of shortened telomeres could mediate breeding age effects in flies and other animals. Shortened telomeres are associated with cellular senescence in some taxa [80], and telomere length can be affected by noncoding telomeric repeat-containing RNAs (TERRA), which are transcriptionally active in *Drosophila* [81]. TERRAs are present in animal (including human) oocytes [82], and in female *Drosophila*, they affect blastoderm formation [83]. Other types of noncoding RNAs could also be involved. Flies maintain chromosome length through retrotranscription [84], which requires complex and

specific chromatin structures [85]. Retrotransposon proliferation can promote mutagenesis [86]. RNA interference (RNAi) mechanisms control the silencing of retrotransposons in germ-line cells [87,88], and parental age effects could be mediated by the transmission of such small noncoding RNAs, with effects on chromatin states and gene expression in embryos [23]. Early development in *Drosophila* is thought to be governed by maternally inherited RNAs and proteins [89], but less is known about the effects of male-derived RNAs on offspring development. Although *T. angusticollis* males do not transmit nutritional nuptial gifts during copulation [90], males probably transfer a variety of microRNAs in the ejaculate. The complement of seminal and egg microRNAs could change with male and female age and affect embryo development.

Another possibility is that flies change their investment in gametes in response to the age or mating experience of their partner. A female may decrease investment per offspring when mated to an older male, whereas a male may reduce the quality or quantity of accessory gland proteins or sperm produced when mated with an older female, resulting in negative effects of parental age on offspring performance. Such responses to mate quality have been reported in *Drosophila* and other insects [91–94] and might be mediated through cuticular hydrocarbons (CHCs) that are known to change with age in flies [95,96]. In our experiment, increasing age was also associated with increasing mating experience. Individuals of both sexes might alter their investment in offspring based on their partner's mating experience, because previously mated males might transfer smaller or lower-quality ejaculates. For example, male mating experience was negatively correlated to nuptial gift quality and sperm number in a bush cricket [97], and female reproductive output was lower when mated with sexually experienced males than when mating with virgin males across 25 species of Lepioptera [98]. Although *T. angusticollis* males appear to be able to replenish their ejaculate reserves very rapidly, the effects of age and mating experience cannot be decoupled statistically in our data and require further investigation.

We quantified effects of ancestors' age at breeding in flies ($F_3$) maintained as virgins in individual containers and supplied with ad libitum food and water. Housing *T. angusticollis* individuals in isolation and as virgins tends to increase their longevity (e.g., the work by Adler and Bonduriansky [99]), whereas ad libitum availability of dietary protein tends to reduce adult longevity [45]. Although our results suggest that larval diet and male competitive environment do not interact strongly with breeding age in affecting longevity of descendants, further work is required to determine whether housing, reproduction, or adult diet of descendants can interact with effects of parental and grand-parental ages at breeding.

Some individuals failed to produce viable offspring or did not survive to breed at older ages, and we cannot exclude the possibility that differential mortality or reproductive success biased the composition of our treatment groups. In particular, because *T. angusticollis* males reared on a nutrient-rich larval diet tend to exhibit an elevated adult mortality rate relative to males reared on a nutrient-poor larval diet [47], fewer $F_1$ focal males from the rich-diet treatment survived to breed at age 35 days, resulting in a smaller sample size for that treatment combination. This resulted in somewhat wider confidence limits for life span and actuarial ageing rate for the $F_3$ descendants of those males, but we cannot exclude the possibility that the elevated $F_1$ mortality was also associated with differential natural selection on males reared on nutrient-rich versus nutrient-poor larval diets.

The interactive effects of grand-parental and parental ages at breeding that we observed suggest that the factors mediating these effects are stable across at least 2 generations. Priest and colleagues [5] suggested that parental age effects could play a role in the evolution of age-ing by contributing to age-related decline in performance and generating selection for earlier reproduction. Bonduriansky and Day [34] argued that if such effects can accumulate over

generations, an environmental change that brings about delayed breeding or causes a more rapid decline in offspring performance with parental age could result in a progressive decline in performance over several generations, resulting in phenotypic changes that resemble the evolution of accelerated ageing. Our results support these ideas by providing experimental evidence that parental age effects can have large effects on descendants' longevity, can occur in both matrilines and patrilines and across contrasting environments, and can be transmitted over at least 2 generations. Further work is needed to understand the context-dependence and fitness consequences of such effects in natural populations.

## Materials and methods

### Source of experimental flies

Experiments were performed using a lab-reared stock of *T. angusticollis* that originated from individuals collected from Fred Hollows Reserve, Randwick, NSW, Australia (33°54′44.04″S 151°14′52.14″E). This stock was maintained as a large, outbred population with overlapping generations and periodically supplemented with wild-caught individuals from the same source population to maintain genetic diversity.

### Larval rearing and diet manipulation

All larvae were reared in climate chambers at 25° C ± 2°C with a 12:12 photoperiod and moistened with deionised water every 2 days. We manipulated the quantity of resources available to larvae during development by rearing flies on either a high-nutrient, standard-nutrient, or low-nutrient larval diet. Diets were based on the work by Sentinella and colleagues [100] and were selected to generate considerable body size differences between treatment groups while minimising larval mortality and to preserve the protein to carbohydrate ratio of approximately 1:3 across diets. All diets consisted of a base of 170 g of cocopeat moistened with 600 mL of reverse osmosis-purified water. The high-nutrient larval diet consisted of 32.8 g of protein (Nature's Way soy protein isolate; Pharm-a-Care, Warriewood, Australia) and 89 g of brown sugar (Woolworths Essentials Bonsucro brand); the standard larval diet consisted of 10.9 g of protein and 29.7 g sugar; the low-nutrient larval diet consisted of 5.5 g of protein and 14.8 g sugar. These nutrients were mixed into the cocopeat and water using a hand-held blender and frozen at −20°C until the day of use. Males and females of the $F_1$ generation were reared on either a high- or low-nutrient larval diet and standardised for larval density (40 eggs per 200 g of larval food). All larvae of the $F_2$ and $F_3$ generations were reared on a standard larval diet (see the work by Adler and colleagues [45] for further details). Following the first adult emergence from each larval container, adult flies were collected for 10 days, and the rest were discarded.

### $F_1$ adult housing and competitive environment

$F_1$ males were subjected to a "low" or "high" competition environment. Each adult focal male was paired with a competitor male reared on a standard larval diet inside an enclosure containing a petri dish with larval medium (which stimulates territory defence behaviours in *T. angusticollis* males). Males in the "high" competition environment were able to move freely around the arena and engage in combat interactions with the competitor male, whereas males in the "low" competition environment were separated by mesh so that they could perceive the competitor's chemical and perhaps visual cues but have no physical contact. All focal $F_1$ females were kept in a similar housing as the "low" competitive environment males where each focal female was paired with a female reared on a standard larval diet. All housing containers had a

layer of moistened cocopeat on the bottom, and dishes of oviposition medium (on which adult flies also feed) to stimulate ovary development in females.

## $F_1$ adult male and female age-at-breeding manipulation

The age at breeding was manipulated for $F_1$ focal individuals by pairing at 'young' (15 ± 1 days old) and 'old' (35 ± 1 days old) ages with an opposite-sex individual reared on the standard larval diet and standardised for age (15 ± 1 days old). These ages were selected because, in *T. angusticollis*, adults become fully reproductively mature by 10 to 15 days of age under laboratory conditions, whereas median longevity of individually housed, captive flies is 37 days for males and 36 days for females, and mortality rate begins to increase rapidly in both sexes after 30 days of age [49]. Thus, at 15 days old, both sexes are considered to be at their prime, whereas, at 35 days old, both sexes are well past their prime. Each focal $F_1$ adult was thus paired twice, each time with a different mate, to produce broods of $F_2$ offspring at 'young' and 'old' ages (Fig 5). Mating pairs were kept in 60 mL glass vials under standardised light and temperature (approximately 23°C) for 1 hour, and females were then placed into 250 mL enclosures with mesh coverings and a moistened cocopeat substrate and were allowed to oviposit for 96 h into a petri dish containing oviposition medium. After 48 h, a fresh oviposition dish was provided. A total of 20 eggs were sampled randomly from each female and transferred to 100 g of standard larval medium.

## $F_2$ adult male and female age-at-breeding manipulation

One $F_2$ male and one $F_2$ female focal individual were randomly sampled for breeding from each $F_1$ larval container. Thus, where possible, each $F_1$ focal individual contributed one $F_2$ offspring of each sex from a reproductive bout at 15 days of age and one $F_2$ offspring of each sex from a reproductive bout at 35 days of age. Each $F_2$ focal individual was paired with a partner of the opposite sex (raised on a standard diet and 15 ± 1 days old on the day of pairing) at 4 ages (where possible): 15 d, 30 d, 45 d, and 60 d. The flies were allowed 1 hour to mate, after which eggs were collected from each female and maintained as described above.

## $F_3$ rearing and quantification of life span

From each reproductive bout of each $F_2$ individual, one male and female of the $F_3$ generation were obtained (where possible) and housed individually in a 120 mL container fitted with a feeding tube containing a sugar-yeast mixture and drinking tube containing water (with both food and water provided ad libitum), and a substrate of moistened cocopeat to maintain humidity. $F_3$ housing containers were maintained at ambient room temperature (23°C ± 4°C) and checked daily for mortality until all individuals had died. To minimise spatial effects, containers were randomly moved to different locations every 2 days.

For all focal individuals, development time and body size were also recorded to investigate their possible roles in mediating treatment effects on life span and mortality rate (refer to S1 and S2 Tables for summary statistics). All $F_1$ and $F_2$ focal individuals were frozen at −20°C after their final reproductive bout (or prior natural death before day 60), and all $F_3$ individuals were frozen after natural death. For all focal $F_1$, $F_2$, and $F_3$ individuals, egg to adult development time was recorded as time from oviposition to adult emergence in days (± 1 day). Thorax length is a reliable proxy for body size in this species [101] and was measured for each $F_1$, $F_2$, and $F_3$ focal individual from images taken using a Leica MS5 stereoscope equipped with a Leica DFC420 digital microscope camera. Measurements were made using FIJI open source software [102].

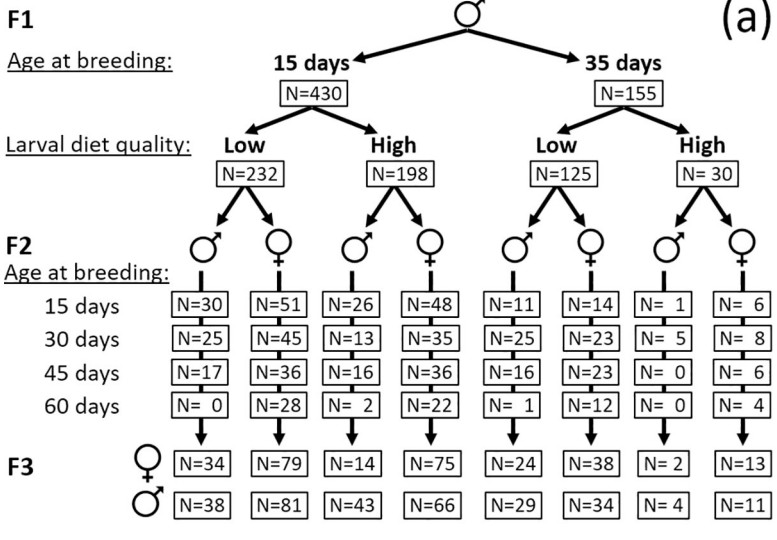

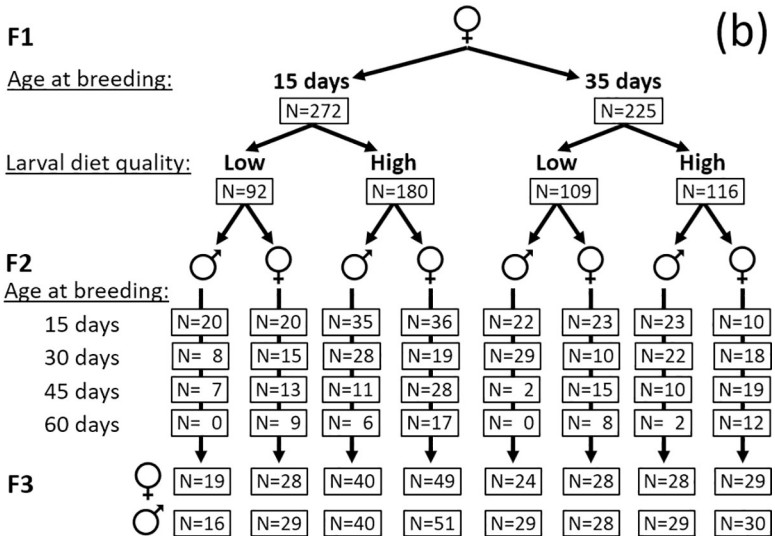

**Fig 5. Experimental design: Patrilines (a) consist of descendants of $F_1$ males, whereas matrilines (b) consist of descendants of $F_1$ females.** $F_1$ individuals were reared on either a high- or low-nutrient larval diet. Adult $F_1$ males were also maintained in high- or low-competition social environments (S4 Table). $F_1$ males and females were then mated at 15 days or 35 days of age, and all offspring ($F_2$) were reared on a standard larval diet. From each $F_1$ breeding bout, 1 male and 1 female of the $F_2$ generation were paired with a standard mate at 15-day intervals up to 60 days of age. Grand-offspring ($F_3$) were all reared on standard larval diet and housed individually until death. Sample sizes (number of $F_1$ or $F_2$ focal individuals that produced offspring and number of $F_3$ individuals for which longevity was quantified) are shown for each combination of treatment and sex. $F_1$, grand-parental generation; $F_2$, female and male offspring; $F_3$, grand-offspring.

## Life span analysis

We investigated treatment effects on $F_3$ life span using R 3.3.2 [103] and the package "lme4" [104]. These analyses facilitate hypothesis testing by making it possible to test interactions within mixed-effects models. Because the life span of every individual was known, no censoring was required. Gaussian linear mixed models (LMM) were used, and all analyses were carried out separately for matrilines (i.e., descendants of $F_1$ females) and patrilines (i.e., descendants of $F_1$ males). Any effects of $F_1$ age at breeding, larval diet, or male competitive

environment therefore represent grand-maternal effects within matrilines and grand-paternal effects within patrilines. Within both matrilines and patrilines, we tested for effects of $F_2$ age at breeding for both female parents (maternal age effects) and male parents (paternal age effects) and compared effects on $F_3$ males and females (i.e., effect of $F_3$ sex). For the patriline data set, $F_1$ male competitive environment and its two-way interactions were tested by a likelihood ratio test (LRT) and were found to have no effect on any dependent variables. The patriline models were then refitted without $F_1$ competitive environment. This resulted in identical model structure for patrilines and matrilines, facilitating comparison of matrilineal and patrilineal results. Qualitatively identical results are obtained without $F_1$ competitive environment as a predictor in the patriline models (S3 Table).

Our final models thus included $F_1$ (grand-parental) larval diet and age at breeding, $F_2$ parental age at breeding, $F_2$ sex and $F_3$ sex as fixed effects. $F_2$ breeding age was fitted as a continuous predictor, whereas the other factors were fitted as categorical predictors. $F_1$ and $F_2$ individual ID, replicate $F_1$ larval container, and emergence date were included as random effects. We also fitted models with $F_1$, $F_2$, and $F_3$ body sizes and development times as fixed covariates in order to determine whether these traits mediate treatment effects on $F_3$ life span (S4 Table). Treatment effects on $F_3$ body size and development time were also tested using similar models to those described above, and results of those analyses are shown in S6 and S7 Figs, S14 and S15 Tables, and discussed in S1 Text. Estimates and F-ratios were obtained using the packages "lme4" [104] and "lmerTest" [105], whereas $p$-values were obtained via "Type 3" likelihood ratio tests using the package "car". To examine the relative effect size of each predictor, we also quantified marginal $R^2$, which is variance explained by fixed factors, and conditional whole model $R^2$ that includes variation explained by random factors from our LMM using the methods developed in [106].

## Mortality rate analysis

To gain a better understanding of treatment effects on $F_3$ life span, we also investigated effects on $F_3$ mortality rates. We used the Bayesian Survival Trajectory Analysis, implemented with the package "BaSTA" [107]. BaSTA utilises a Bayesian approach based on Markov Chain Monte Carlo (MCMC) estimation of age-specific mortality rate distributions. Our data are uncensored, and the date of adult emergence is known for all individuals, allowing us to obtain reliable population estimates of the mortality distribution [108]. In order to find the mortality rate distribution that best fits our data, we first used the package "flexsurv" [109] on a combined data set comprising both patrilines and matrilines. We compared the simple and Makeham versions of the Gompertz and Weibull models, as well as the logistic and exponential models, using the Akaike Information Criterion (AIC). This analysis showed that a simple Gompertz distribution provided the best fit to our data (S5 Table). Mortality rate was therefore modeled as

$$\mu_b(x|b) = e^{b_0 + b_1 x}.$$

Survival probability was modeled as

$$S_b(x|b) = exp\left[\frac{e^{b_0}}{b_1}(1 - e^{b_1 x})\right].$$

The Gompertz mortality rate function includes a scale parameter, $b_0$ (often called the "baseline mortality rate"), and a shape parameter, $b_1$, that describes the dependency of mortality on age ($x$) and is often interpreted as the rate of actuarial ageing, which reflects the rate of increase in mortality rate with age [110–113].

We used BaSTA to estimate and compare parameters of the simple Gompertz model for our experimental treatment groups. We performed 4 parallel BaSTA simulations, each proceeding for 2,200,000 iterations, with a burn-in of 200,000 chains, and took an MCMC chain sample every 4,000 iterations. Our models generated parameter estimates that converged with low serial autocorrelations (<5%) and robust posterior distributions of $b_o$ and $b_1$ ($N = 2,000$), allowing for robust comparisons between treatment groups.

We compared parameter estimates for various treatment groups based on differences between their posterior distributions, using the KLDC implemented in BaSTA. Values near 0.5 suggest nominal differences between distributions, whereas values close to 1 indicate a sizeable divergence. KLDC thresholds can vary depending on interpretation and can range between 0.65 and 1 [114–116]. We considered a relatively conservative KLDC value >0.85 to indicate a difference between the posterior distributions of the treatment groups being compared. We report Gompertz $b_o$ parameter estimates on a log scale and refer to $F_1$ treatment combinations as HO, HY, LO, and LY.

Data are deposited in the Dryad repository: https://doi.org/10.5061/dryad.2rbnzs7hw [117].

## Supporting information

**S1 Fig. Combined effects of $F_1$, $F_2$ breeding ages and $F_1$ larval diet quality on mean $F_3$ life span.** Black lines represent $F_3$ individuals descended from $F_1$ males and females bred at a young age (15 days old) and red lines signify individuals descended from old (35 days old) grandparents. In patrilines only, individuals descended from $F_1$ males that were subjected to either a high or low competitive environment are represented by a solid or dotted line, respectively. $F_3$ grand-offspring of $F_1$ grandparents reared on a high-nutrient larval diet are represented by a circle, and low-nutrient larval diest is represented by a triangle. All points represent means. Bars represent SEM. Underlying data can be found in the Dryad Repository: https://doi.org/10.5061/dryad.2rbnzs7hw. F1, grand-parental generation; F2, female and male offspring; F3, grand-offspring.
(TIF)

**S2 Fig. Values of the KLDC for patrilines, comparing parameter posterior distributions between treatment groups.** Underlying data can be found in the Dryad Repository: https://doi.org/10.5061/dryad.2rbnzs7hw. HO, High Nutrient Old Breeding treatment; HY, High Nutrient Young Breeding treatment; KLDC, Kullback-Leibler discrepancy calibration; LO, Low Nutrient Old Breeding treatment; LY, Low Nutrient Young Breeding treatment.
(TIF)

**S3 Fig. Values of the KLDC for matrilines, comparing parameter posterior distributions between our treatment groups.** Underlying data can be found in the Dryad Repository: https://doi.org/10.5061/dryad.2rbnzs7hw. HO, High Nutrient Old Breeding treatment; HY, High Nutrient Young Breeding treatment; KLDC, Kullback-Leibler discrepancy calibration; LO, Low Nutrient Old Breeding treatment; LY, Low Nutrient Young Breeding treatment.
(TIF)

**S4 Fig. Values of the KLDC for patrilines, comparing parameter posterior distributions between treatment groups.** Underlying data can be found in the Dryad Repository: https://doi.org/10.5061/dryad.2rbnzs7hw. $F_1$, grand-parental generation; $F_2$, female and male offspring; KLDC, Kullback-Leibler discrepancy calibration; OO, Old $F_1$ breeding age Old $F_2$ breeding age; OY, Old $F_1$ breeding age Young $F_2$ breeding age; YO, Young $F_1$ breeding age Old $F_2$ breeding age treatment; YVO, Young $F_1$ breeding age Very old $F_2$ breeding age; YY,

Young $F_1$ breeding age Young $F_2$ breeding age.
(TIF)

**S5 Fig. Values of the KLDC for matrilines, comparing parameter posterior distributions between our treatment groups.** Underlying data can be found in the Dryad Repository: https://doi.org/10.5061/dryad.2rbnzs7hw. $F_1$, grand-parental generation; $F_2$, female and male offspring; KLDC, Kullback-Leibler discrepancy calibration; OO, Old $F_1$ breeding age Old $F_2$ breeding age; OY, Old $F_1$ breeding age Young $F_2$ breeding age; YO, Young $F_1$ breeding age Old $F_2$ breeding age treatment; YVO, Young $F_1$ breeding age Very old $F_2$ breeding age; YY, Young $F_1$ breeding age Young $F_2$ breeding age.
(TIF)

**S6 Fig. Effects of $F_2$ breeding age and $F_2$ sex on $F_3$ body size in patrilines.** Solid grey lines represent $F_3$ individuals descended from $F_2$ females and solid black lines represent $F_3$ individuals descended from $F_2$ males. Bars represent SEM. Underlying data can be found in the Dryad Repository: https://doi.org/10.5061/dryad.2rbnzs7hw. $F_2$, female and male offspring; $F_3$, grand-offspring.
(TIF)

**S7 Fig. Effects of $F_1$ larval diet and age at breeding on $F_3$ body size in patrilines and matrilines.** Solid grey lines represent effects of $F_1$ individuals reared on reared on a poor larval diet, and solid black lines represent the effects of $F_1$ individuals reared on a rich larval diet. Bars represent SEM. Underlying data can be found in the Dryad Repository: https://doi.org/10.5061/dryad.2rbnzs7hw. $F_1$, grand-parental generation; $F_3$, grand-offspring.
(TIF)

**S1 Table. Factorial summary of mean $F_3$ life span, development time, and thorax length for patrilines.** $F_3$, grand-offspring.
(XLSX)

**S2 Table. Factorial summary of mean $F_3$ life span, development time, and thorax length for matrilines.** $F_3$, grand-offspring.
(XLSX)

**S3 Table. Linear mixed-effects models of $F_3$ life span for patrilines including $F_1$ competitive environment.** Negative effects for $F_1$ larval diet indicate that grandparents reared on a high-nutrient larval diet produced grand-offspring with a relatively longer life span than descendants of grandparents reared on a low-nutrient larval diet. Negative effects of $F_1$ and $F_2$ age indicate that old grandparents and parents produced $F_3$ individuals with reduced lifespans, negative effects of larval diet indicate that low-nutrient larval diet has a negative effect on $F_3$ life span, and negative effects of sex indicate that the life span of male descendants was lower than that of females. Significance codes: p = 0.0001 '***', p = 0.001 '**', p = 0.01 '*', p = 0.05 '.', p = 0.1. $F_1$, grand-parental generation; $F_2$, female and male offspring; $F_3$, grand-offspring
(XLSX)

**S4 Table. Linear mixed-effects models of $F_3$ life span for patrilines and matrilines, with thorax length and development time of all focal individuals included as covariates.** Negative effects for $F_1$ larval diet indicate that grandparents reared on a high-nutrient larval diet produced grand-offspring with a relatively longer life span than descendants of grandparents reared on a low-nutrient larval diet. Negative effects of $F_1$ and $F_2$ age indicate that old grandparents and parents produced $F_3$ individuals with reduced lifespans, negative effects of larval diet indicate that low-nutrient larval diet has a negative effect on $F_3$ life span, and negative

effects of sex indicate that the life span of male descendants was lower than that of females. Significance codes: p = 0.0001 '***', p = 0.001 '**', p = 0.01 '*', p = 0.05 '.', p = 0.1. $F_1$, grand-parental generation; $F_2$, female and male offspring; $F_3$, grand-offspring
(XLSX)

**S5 Table. Model selection results based on 'flexsurv' package.** The simple Gompertz model provided the best fit to our data based on the Aikaike Information Criterion and was used for further analyses using BaSTA.
(XLSX)

**S6 Table. Parameter estimates for each treatment group for the best fitting model (Gompertz with simple shape) for grand-paternal effects of $F_1$ larval diet × $F_1$ breeding age.** $F_1$, grand-parental generation
(XLSX)

**S7 Table. Mean KLDC values for patrilines, comparing parameter posterior distributions between $F_1$ treatment groups.** $F_1$, grand-parental generation; KLDC, Kullback-Leibler discrepancy calibration
(XLSX)

**S8 Table. Parameter estimates for each treatment group for the best fitting model (Gompertz with Simple shape) for grand-maternal effects of $F_1$ larval diet × $F_1$ breeding age.** $F_1$, grand-parental generation
(XLSX)

**S9 Table. Mean KLDC values for matrilines, comparing parameter posterior distributions between $F_1$ treatment groups.** $F_1$, grand-parental generation; KLDC, Kullback-Leibler discrepancy calibration
(XLSX)

**S10 Table. Parameter estimates for each treatment group for the best fitting model (Gompertz with simple shape) for effects of $F_1$ breeding age × $F_2$ breeding age in patrilines.** $F_1$, grand-parental generation; $F_2$, female and male offspring
(XLSX)

**S11 Table. Mean KLDC values for patrilines comparing parameter posterior distributions between treatment groups.** KLDC, Kullback-Leibler discrepancy calibration
(XLSX)

**S12 Table. Parameter estimates for each treatment group for the best fitting model (Gompertz with Simple shape) for effects of $F_1$ breeding age × $F_2$ breeding age in matrilines.** $F_1$, grand-parental generation; $F_2$, female and male offspring
(XLSX)

**S13 Table. Mean KLDC values for matrilines, comparing parameter posterior distributions between our treatment groups.** KLDC, Kullback-Leibler discrepancy calibration
(XLSX)

**S14 Table. Linear mixed-effects model of $F_3$ body size. Significant effects are highlighted in bold.** Significance codes: $p = 0.0001$ '***', $p = 0.001$ '**', $p = 0.01$ '*', $p = 0.05$ '.', $p = 0.1$. $F_3$, grand-offspring
(XLSX)

**S15 Table. Linear mixed-effects model of F₃ development time.** Significant effects are highlighted in bold. Solid black lines represent the development time of $F_3$ offspring descended from $F_2$ males, and solid grey lines are individuals derived from $F_2$ females. Bars represent SEM. SEM for descendants of $F_2$ females at 60 days is missing because of a small sample size. Significance codes: $p = 0.0001$ '***', $p = 0.001$ '**', $p = 0.01$ '*', $p = 0.05$ '.', $p = 0.1$.$F_1$, grandparental generation; $F_2$, female and male offspring; $F_3$, grand-offspring
(XLSX)

**S1 Text. Discussion of the effects influencing F₃ body size and development time.** $F_3$, grand-offspring
(DOCX)

## Acknowledgments

The authors would thank E. Macartney, N. W. Burke, and F. Zajitschek for their helpful comments and discussions of earlier drafts of the manuscript.

## Author Contributions

**Conceptualization:** Zachariah Wylde, Amy K. Hooper, Russell Bonduriansky.

**Data curation:** Zachariah Wylde, Foteini Spagopoulou, Amy K. Hooper.

**Formal analysis:** Zachariah Wylde, Foteini Spagopoulou.

**Funding acquisition:** Russell Bonduriansky.

**Investigation:** Zachariah Wylde.

**Methodology:** Foteini Spagopoulou, Amy K. Hooper, Russell Bonduriansky.

**Project administration:** Russell Bonduriansky.

**Resources:** Russell Bonduriansky.

**Supervision:** Alexei A. Maklakov, Russell Bonduriansky.

**Validation:** Zachariah Wylde, Russell Bonduriansky.

**Writing – original draft:** Zachariah Wylde, Russell Bonduriansky.

**Writing – review & editing:** Zachariah Wylde, Foteini Spagopoulou, Amy K. Hooper, Alexei A. Maklakov, Russell Bonduriansky.

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
