## [Editor Report · Decision Letter 0]

1 Aug 2019

Dear Dr Wylde, 

Thank you for submitting your manuscript entitled "Parental breeding age effects on descendants’ longevity interact over two generations in matrilines and patrilines" for consideration as a Research Article by PLOS Biology.

Your manuscript has now been evaluated by the PLOS Biology editorial staff as well as by an academic editor with relevant expertise and I am writing to let you know that we would like to send your submission out for external peer review.

*Please be aware that, due to the voluntary nature of our reviewers and academic editors, manuscripts may be subject to delays during the holiday season. Thank you for your patience.*

Please re-submit your manuscript within two working days, i.e. by Aug 03 2019 11:59PM.

Kind regards,

Di Jiang, PhD

Associate Editor

PLOS Biology

---

## [Decision Letter · Decision Letter 1]

9 Sep 2019

Dear Dr Wylde,

Thank you very much for submitting your manuscript "Parental breeding age effects on descendants’ longevity interact over two generations in matrilines and patrilines" for consideration as a Research Article at PLOS Biology. Your manuscript has been evaluated by the PLOS Biology editors, an Academic Editor with relevant expertise, and by four independent reviewers.

In light of the reviews (below), we are pleased to offer you the opportunity to address all the comments from the reviewers in a revised version that we anticipate should not take you very long. We will then assess your revised manuscript and your response to the reviewers' comments and we may consult the reviewers again.

Your revisions should address the specific points made by each reviewer. Please submit a file detailing your responses to the editorial requests and a point-by-point response to all of the reviewers' comments that indicates the changes you have made to the manuscript. In addition to a clean copy of the manuscript, please upload a 'track-changes' version of your manuscript that specifies the edits made. This should be uploaded as a "Related" file type. You should also cite any additional relevant literature that has been published since the original submission and mention any additional citations in your response. 

Before you revise your manuscript, please review the following PLOS policy and formatting requirements checklist PDF: http://journals.plos.org/plosbiology/s/file?id=9411/plos-biology-formatting-checklist.pdf. It is helpful if you format your revision according to our requirements - should your paper subsequently be accepted, this will save time at the acceptance stage.

Please note that as a condition of publication PLOS' data policy (http://journals.plos.org/plosbiology/s/data-availability) requires that you make available all data used to draw the conclusions arrived at in your manuscript. If you have not already done so, you must include any data used in your manuscript either in appropriate repositories, within the body of the manuscript, or as supporting information (N.B. this includes any numerical values that were used to generate graphs, histograms etc.). For an example see here: http://www.plosbiology.org/article/info%3Adoi%2F10.1371%2Fjournal.pbio.1001908#s5.

For manuscripts submitted on or after 1st July 2019, we require the original, uncropped and minimally adjusted images supporting all blot and gel results reported in an article's figures or Supporting Information files. We will require these files before a manuscript can be accepted so please prepare them now, if you have not already uploaded them. Please carefully read our guidelines for how to prepare and upload this data: https://journals.plos.org/plosbiology/s/figures#loc-blot-and-gel-reporting-requirements.

Upon resubmission, the editors assess your revision and assuming the editors and Academic Editor feel that the revised manuscript remains appropriate for the journal, we may send the manuscript for re-review. We aim to consult the same Academic Editor and reviewers for revised manuscripts but may consult others if needed.

We expect to receive your revised manuscript within one month. Please email us (plosbiology@plos.org) to discuss this if you have any questions or concerns, or would like to request an extension. At this stage, your manuscript remains formally under active consideration at our journal; please notify us by email if you do not wish to submit a revision and instead wish to pursue publication elsewhere, so that we may end consideration of the manuscript at PLOS Biology.

When you are ready to submit a revised version of your manuscript, please go to https://www.editorialmanager.com/pbiology/ and log in as an Author. Click the link labelled 'Submissions Needing Revision' where you will find your submission record. 

Sincerely,

Di Jiang, PhD

Associate Editor

PLOS Biology

Reviewer remarks:

Reviewer #1: Wylde and co-authors present a well-motivated and thorough examination of parental age effects in longevity in neriid flies. The authors do an excellent job of placing their study in broader context, and pointing to various avenues of further work that might help to clarify the role of parental age in the evolution of ageing, as well as the potential mechanisms. These issues are currently neglected and I think the authors' contribution will be a strong addition to the ageing literature. The statistical analyses are certainly sufficient, although presentation / interpretation of the lme4 models need a little work (detailed below). My comments are therefore fairly minor, centred mostly around clarifying a few aspects, improving presentation of results, and a suggestion for ending on a stronger note than the current manuscript.

79-82: consider dropping “we investigated” / “we investigated and compared” from each bullet point, particularly as “we addressed” is in the opening clause. This redundancy makes this part long and difficult to digest when it should be a punchy section about the authors’ aims.

92-94: a note of lifespan, patterns of reproductive senescence etc from previous studies of these flies would be useful here to show the reader why these ages have been chosen.

Table 1: For categorical variables, we need to know what the reference levels / contrasts are, else having a table of parameter estimates isn’t particularly meaningful. I’m also wondering why the authors use likelihood ratio tests to assess significance of covariates, but then use p-values from F-tests in their table to illustrate significant effects? It seems to me that a more coherent setup would be to include estimate, SE and F-statistic for each term in the table, then include chi-square statistic and associated p-value for interactions / main effects that are not involved in interactions. For an example, see Table 2 of Buser et al 2013 Fun Ecol (DOI: 10.1111/1365-2435.12188).

292-302: Please clarify the abbreviations for treatment groups somewhere in here. Also, consider including the confidence intervals for each estimate (or at least noting that the posterior distributions are shown in Fig 4). While on the subject, I’m not sure the authors discuss the fact that high-nutrient old age groups seem to have much higher variance in both survival and mortality than all other groups. Is there any reason that that might be the case?

Figure 4: The figure text (axes labels etc) is barely readable despite being a full page – please enlarge this. Furthermore, the legend doesn’t adequately describe the figure panels – this needs to be clear to the reader that the left hand side shows effect sizes and (95%?) posterior distributions for the 4 treatment groups, and that the right hand side shows how these estimates translate to survival and mortality rates over time. In addition, the x axis states that age is in years – I assume this is a typo, otherwise I must congratulate the authors even more on the completion of this study!

344-348: I don’t quite get how the ‘indeed’ clause follows what comes before it. Is there a step in the logic missing here? Or it just needs to be rephrased somehow to make it clear what the authors mean.

443-452: I think the authors do themselves a little bit of a disservice with their final paragraph, and could make a stronger case for the usefulness of their results in the broader picture. Currently it feels like the paper fades to ‘something was seen, and more work should be done’. This might even just be a case of rewriting the first sentence in active voice rather than passive, but I think the last line could also be reworked to make the case that the authors have indeed contributed results that demonstrate the need for further work in a neglected aspect of the evolution of ageing.

Reviewer #2: This MS investigates the impact of age on aging in offspring and grand-offspring in a fly. It's major advance is that it comprehensively investigates both male and female age on the aging of descendants. They find that male age has at least as big an impact as female age, both at the parental and grandparental stage, and the resulting lifespan change in descendants is very impressive (up to 40%). In my view, this paper is a substantial advance in the field, and will become a citation classic. So I am happy to recommend it for publication. It is very well written and comprehensively covers the literature, and I have only very minor suggestions for improvement. 

Minor issues:

The authors briefly discuss the impact of females potentially investing less in the offspring of older males, but they do not mention the possibility that mating experience (by both males and females) correlates with age in their design. This also might change investment by F1 and F2 flies in ejaculates and eggs. This is not really a flaw in their design, but I think it is worth a sentence or two.

Figure 1 doesn't seem to reflect the design. My understanding is that the 15 and 35 day old broods from the F1 became the F2, who were reared under standard conditions and mated at age 15, 30, 45 and 60. The current diagram does not depict that. I cannot see why the F2 flies have a branching arrow beneath them- what are the F3 flies being split into two groups for? Instead, a simple single arrow point down from each of the F2 male/female symbols would make more sense (ie twice as many arrows as currently used, but none of them split at the end).

Reviewer #3: This manuscript reports the results of an experiment testing the possible effect of grandmaternal and grandpaternal age on offspring lifespan. The article is well written and reports interesting results. However, I have a few major criticisms.

The first point is that, by reading the manuscript, one might misleadingly think that paternal age effects on the progeny have not been studied, yet. This is not true since during the last years several papers have focused on paternal age and have already reported negative effects on offspring lifespan and/or LRS (see for instance Schroeder et al. 2015 PNAS). I think that this should be fully acknowledged. 

I found the experimental design unnecessary complex. Looking at how the environment modulates grandparental effects on offspring longevity is of course interesting, but to me the first step should have been to make sure that such grandparental effects exist, using a more straightforward experimental design. The choice of the environmental traits that have been manipulated is also questionable, especially male’s competitive environment, given that previous work showed no effect of male combat on reproductive aging in this species.

The choice of the age at breeding for the different generations should also be better explained and justified. Grandparents were let to breed at the age of 15 and 35 days, but I did not find any justification of these ages and why 15 day old flies are young and 35 day old flies are old. This point is crucial to me especially because at the following generation flies were bred at the age of 15, 30, 45 and 60 days. Therefore, 35 day old flies seem to be middle-aged rather than old. I think the authors should provide data on the onset of reproductive aging, as to justify that 35 day old flies are indeed “old”. The other concern is that males and females were bred at the same age but there is evidence showing that the onset and rate of aging might differ between sexes. Is the onset of reproductive aging similar between sexes in this species? If not the comparison of grandmaternal and grandpaternal effects on offspring longevity might be flawed.

The final point related to the experimental design is that focal flies (the F3 generation) were maintained in isolation and therefore could not breed. This is of course a very artificial condition, very different from what these animals experience in the wild. This brings me to the point of the ecological relevance of the experiment. As also acknowledged by the authors in the discussion, flies have a much longer longevity in the lab compared to the wild, therefore we can reasonably question whether the results reported here have any ecological relevance.

Statistical analyses

I did not find any mention of the sample size, except the overall number of male and female flies per generation reported in figure 1. Looking at figure 2, it seems that much fewer F3 flies were available in the F1 high nutrient diet x old breeding age group. Why is this so? I would recommend reporting more precisely the number of flies per group. 

Table 1. I do not understand if this table reports the results of the full model or if there have been a model selection procedure. If not, I think this should be done because many of the effects (especially the interaction terms) seem to be borderline (p values between 0.01 and 0.05). I would also recommend providing an estimate of effect size. 

Lifespan was analyzed using a normal distribution of errors, but very often longevity is right skewed with a few individuals having extreme lifespan. Did you check the assumptions underlying the use of LMM?

Figure 2. This might be a matter of taste, but I do not think “violins” are particularly useful here. I am also wondering why you decided to split the data according to F1 breeding age and diet, given that diet is never significant. Instead, I would recommend illustrating how good was your model to fit the data. 

Figure 3. Sample sizes should be reported here. How many flies were in the group 35 day old F1 x 60 day old F2.

Reviewer #4: Wylde et al., ‘Parental breeding age effects on descendants’ longevity interact over two generations in matrilines and patrilines’, PBIOLOGY-D-19-02204 

This is a potentially important paper assessing parental and grand-parental breeding age effects on lifespan in the neriid fly Telostylinus angusticollis. The study not only takes parental and grand-parental age effects along the patriline and matriline into account but also looks at interactive - potentially cumulative - effects across generations and addresses whether inter-/transgenerational age effects are sensitive to environmental modifiers (diet, male competitive behaviors). 

The key findings reported are that older breeding ages along the patriline and matriline are associated with negative effects on lifespan in offspring (which is in line with findings in other species) and the authors also provide some evidence for cumulative effects of advanced grand-parental and parental ages on lifespan in progeny (this aspect has not been addressed by prior studies). Parental/grand-parental age effects were not modified by the environmental factors studied. 

Overall, the study appears to be well-designed, takes an extensive approach considering a number of possible parental/grand-parental factors and may represent an important contribution to the existing literature on multi-generational age effects. 

This reviewer has the following specific comments on the manuscript:

1) It would be important to specify for each of the different breeding age groups what proportion of animals in the F1/F2 population died prior to reaching the assigned breeding age. Also, do the authors have data regarding possible age-related changes in breeding success? These aspects are important because of selection processes that are expected to affect the various breeder age groups in different ways. 

2) Please clarify how lifespans measured in the present study compare to published lifespan data in Telostylinus angusticollis. 

3) Page 7, lines 145-165: The same F1/F2 animals were used at different ages to generate offspring of the various breeder age groups. Is it correct that, as a consequence, young breeder offspring were always generated by naïve animals and older breeder offspring were always generated by experienced breeders? In that case, it would be essential to make sure that indeed breeder age is the critical experimental variable (and not parity/prior breeding experience). Do the authors have offspring lifespan data derived from naïve breeders of the various parental/grand-parental age groups?

4) Fig. 1: It is essential to spell out how many animals precisely were used within all the different experimental groups covered by the study design and how many breeders they were derived from. The scheme in Fig. 1 should be modified to represent a tree showing all experimental groups. All possible combinations of factors examined and the number of associated animals and breeders should be presented within this scheme. 

5) Fig. 2 and Fig. 3: These figures show data that were stratified by some of the factors used in the present study but were collapsed across others. While it may be useful to simplify data presentation in the main figures, it would still be informative to provide additional figures - within the Supplementary Material - wherein the data presented in Figs. 2 and 3 are stratified by all the relevant factors that went into the analysis (sex, diet, competition etc., as applicable).

6) For all measures examined (lifespan, thorax length etc.), please provide tables specifying key data distribution metrics (mean, standard deviation; also, number of animals, number of breeders) for all the different groups/combination of factors (see also point 4 above). This would be an essential addition and could be placed in the Supplementary Material.

7) Please check reference 37 (Sharma et al., 2016) – it does not appear to support the statement on page 4, lines 57-59 of the manuscript.

---

## [Editor Report · Decision Letter 2]

22 Oct 2019

Dear Dr Wylde,

Thank you for submitting your revised Research Article entitled "Parental breeding age effects on descendants’ longevity interact over two generations in matrilines and patrilines" for publication in PLOS Biology. I have now obtained advice from the Academic Editor who has assessed your revision. 

We're delighted to let you know that we're now editorially satisfied with your manuscript. However before we can formally accept your paper and consider it "in press", we also need to ensure that your article conforms to our guidelines; several of which are described below and are marked with "***IMPORTANT: ". A member of our team will be in touch shortly with a set of requests. As we can't proceed until these requirements are met, your swift response will help prevent delays to publication.

Please note that you may have the opportunity to make the peer review history publicly available. The record will include editor decision letters (with reviews) and your responses to reviewer comments. If eligible, we will contact you to opt in or out.

Sincerely,

Di Jiang

PLOS Biology

FINANCIAL DISCLOSURE and COMPETING INTERESTS:

You write, in the submission form: 

"https://www.arc.gov.au/

Future Fellowship FT120100274 and Discovery Grant DP170102449

Awarded to Professor Russell Bonduriansky. The funder had an integral role to the study design, analysis, decision to publish and the preparation of the manuscript."

***IMPORTANT: Please confirm if the funder indeed had 'an integral role to the study design, analysis, decision to publish and the preparation of the manuscript'. And if so, please let us know if the funder's involvement constitutes a competing interest. 

DATA POLICY:

***IMPORTANT: Regardless of the method selected, please ensure that you provide the individual numerical values that underlie the summary data displayed in the following figure panels: 2, 3, 4, 5, S1, S2, S3, S4, S5, S6, S7, as they are essential for readers to assess your analysis and to reproduce it. ***IMPORTANT: Please also ensure that figure legends in your manuscript include information on where the underlying data can be found. You can write, in every relevant figure legend, that, e.g., "Underlying data are found in S1 Data."

***IMPORTANT: You declare in the submission form that "All data files are available from datadryad.org database DOI: https://doi.org/10.5061/dryad.1b6p398". Please make sure that these files are available to us to check, and if needed, please provide a reviewer key/token for us to access the files. Currently, the website says: 'DOI Not Found'.

---

## [Editor Report · Decision Letter 3]

7 Nov 2019

Dear Dr Wylde,

On behalf of my colleagues and the Academic Editor, Nick H. Barton, I am pleased to inform you that we will be delighted to publish your Research Article in PLOS Biology. 

Early Version

PRESS 

Kind regards,

Sofia Vickers

Senior Publications Assistant

PLOS Biology

On behalf of, 

Di Jiang,

Associate Editor

PLOS Biology